# A Brain-Targeted Approach to Ameliorate Memory Disorders in a Sporadic Alzheimer’s Disease Mouse Model via Intranasal Luteolin-Loaded Nanobilosomes

**DOI:** 10.3390/pharmaceutics14030576

**Published:** 2022-03-05

**Authors:** Manal A. Elsheikh, Yasmin A. El-Feky, Majid Mohammad Al-Sawahli, Merhan E. Ali, Ahmed M. Fayez, Haidy Abbas

**Affiliations:** 1Department of Pharmaceutics, Faculty of Pharmacy, Damanhour University, Damanhour 22511, Egypt; 2Department of Pharmaceutics, Faculty of Pharmacy, Modern University for Technology and Information (MTI), Cairo 11571, Egypt; yasmin.elfeky@pharm.mti.edu.eg; 3Department of Pharmaceutical Technology, Faculty of Pharmacy, Kafr Elsheikh University, Kafr Elsheikh 33516, Egypt; majid.alsawahli@gu.edu.eg; 4Department of Pharmaceutics, College of Pharmacy, The Islamic University, Najaf 54001, Iraq; 5Department of Pathology, Faculty of Veterinary Medicine, Cairo University, Giza 12211, Egypt; merhan.essam@vet.cu.edu.eg; 6Department of Pharmacology and Toxicology, School of Life and Medical Sciences, University of Hertfordshire Hosted by Global Academic Foundation, New Administrative Capital, Cairo 11835, Egypt; a.fayez@gaf.edu.eg

**Keywords:** luteolin, bile salts, nanovesicles, factorial design, sporadic Alzheimer’s disease, memory dysfunction, brain targeting, intranasal drug delivery

## Abstract

Impaired memory and cognitive function are the main features of Alzheimer’s disease (AD). Unfortunately, currently available treatments cannot cure or delay AD progression. Moreover, the blood–brain barrier hampers effective delivery of treatment to the brain. Therefore, we aimed to evaluate the impact of intranasally delivered luteolin on AD using bile-salt-based nano-vesicles (bilosomes). Different bilosomes were prepared using 2^3^-factorial design. The variables were defined by the concentration of surfactant, the molar ratio of cholesterol:phospholipid, and the concentration of bile salt. Results demonstrated optimized luteolin-loaded bilosomes with particle size (153.2 ± 0.98 nm), zeta potential (−42.8 ± 0.24 mV), entrapment efficiency% (70.4 ± 0.77%), and % drug released after 8 h (80.0 ± 1.10%). In vivo experiments were conducted on an AD mouse model via intracerebroventricular injection of 3 mg/kg streptozotocin. We conducted behavioral, biochemical marker, histological, and immune histochemistry assays after administering a luteolin suspension or luteolin bilosomes (50 mg/kg) intranasally for 21 consecutive days. Luteolin bilosomes improved short-term and long-term spatial memory. They also exhibited antioxidant properties and reduced levels of proinflammatory mediators. They also suppressed both amyloid β aggregation and hyperphosphorylated Tau protein levels in the hippocampus. In conclusion, luteolin bilosomes are an effective, safe, and non-invasive approach with superior cognitive function capabilities compared to luteolin suspension.

## 1. Introduction

Globally, patients with neurodegenerative disorders, such as Parkinson’s disease, schizophrenia, migraine, and Alzheimer’s disease (AD), have limited treatment options. AD is one of the fatal neurodegenerative diseases characterized by advanced memory and cognition deterioration [1]. Most AD cases are determined to be sporadic AD (SAD) and are typically diagnosed in patients aged 65 or older [2]. β-Amyloid (Aβ) plaque aggregation and deposition is most likely the main driver of AD pathogenesis. However, recent evidence indicates that many other factors, such as oxidative stress and neuroinflammation, also contribute to AD progression [3]. Unfortunately, to date, the available pharmacological treatments are limited and cannot cure AD or delay its progression. These include symptomatic treatments using FDA-approved anticholinesterase medications (donepezil, galantamine, tacrine, and rivastigmine) [4]. However, such treatments can induce side effects, such as headache, dizziness, nausea, vomiting, and appetite loss [4]. Other treatment strategies, such as antioxidants and anti-inflammatory agents, may be useful against memory-related disorders [4]. However, many hurdles can hamper drug targeting to the brain, such as the blood–brain barrier (BBB), which blocks most neurological drugs administered either orally or via the invasive parental route, leading to side effects, as the drugs can affect non-targeted organs [5,6]. There is a high demand for targeted and effective drug delivery to the site of action in the organism [7]. Among them, the development of non-invasive drug delivery to the brain for treatment of chronic central nervous system disorders is challenging. The olfactory route is an excellent alternative to the oral or parental routes regarding brain targeting since it allows the administration of lower doses with minimum off-target side effects [8,9]. Furthermore, intranasal (I.N) drug delivery through the olfactory pathway offers many advantages: it is non-invasive, bypasses the BBB and hepatic/intestinal enzymatic degradation, and prevents systemic absorption, thereby preventing toxicity to other organs [4,6,10]. Consequently, the use of a controlled-release nanoplatform via the olfactory mucosa may be a promising tool in the management of AD.

Recently, worldwide pharmaceutical nanotechnology researchers have focused on nanocarriers with the ability to simulate biological components. Among these, bile-salt-based nanovesicles have recently been recognized as a prominent tactic to improve the in vivo efficacy of conventional vesicular carrier systems [11]. Bile salts are endogenous surfactants that possess powerful solubilization and emulsification characteristics. Pharmaceutically, they are used as solubility enhancers for hydrophobic drugs and permeation enhancers across biological membranes [12,13]. Integrating bile salts within the lipid bilayer of vesicular carriers forms self-assembling structures called “bilosomes” [14,15]. Bilosomes (BLs) are more chemically and physiologically stable compared to other vesicular nanocarriers (such as liposomes, emulsomes, niosomes, and transferosomes) and require no special storage conditions [9,16].

“Food-rich phytochemicals” have recently gained enormous interest in pharmaceutical research globally, in particular, in the field of nanomedicine. Such phytoceuticals exhibit lower toxicity and a broad range of therapeutic activities. Among them, luteolin (LUT, Appendix A) is a yellow flavonoid found in various types of food, such as parsley, celery, oregano, peppermint, and thyme. Several studies have reported that LUT has potential pharmacological effects, including anticancer, antioxidant, and anti-inflammatory activities [17]. Additionally, LUT possesses neuroprotective activity, as it can suppress Aβ deposition, down-regulates the expression of oxidative stress markers (by increasing glutathione levels and scavenging reactive oxygen species), and reduces levels of proinflammatory mediators (NOS, COX-2, and TNF-α) [18]. However, its poor water solubility, extensive first-pass metabolism, and poor BBB permeability prevent its oral delivery [19,20].

Many attempts have been made in the literature to enhance LUT solubility and dissolution rate [21]. These include salt formation; prodrug analogs; nanosolubilization through particle size reduction (e.g., nanosuspension, nanoemulsion) [22]; and complex formation with surfactants, phospholipids, or cyclodextrins [17,23,24,25]. However, the enhancement of LUT permeability through BBB and brain targeting has not yet been implemented in the literature.

Therefore, the current investigation developed LUT-loaded BLs as a novel tool to enhance LUT solubility and BBB permeability and circumvent its first-pass metabolism. This study is considered the first one to evaluate brain targeting of luteolin via intranasal bilosomes through the olfactory mucosa. A 2^3^ full factorial design was adopted to investigate different formulation variables [26,27]. We evaluated the influence of formulation variables on vesicle size, entrapment efficiency, zeta potential, and % drug released. Furthermore, biological evaluation was performed to assess the potential of brain delivery after intranasal administration of LT-loaded BLs. In conclusion, the hypothesized formulation could be a promising tool for the management of AD with high efficacy and rapid onset of action followed by an extended drug release.

## 2. Materials and Methods

Luteolin ≥98% (LT) was supplied by Baoji Guokang Bio-Technology Co. (Baoji, China). Lipoid^®^S100 (LP) l-α-phosphatidylcholine, M.wt = 786.1 g/mol was obtained from Lipoid AG (Ludwigshafen, Germany). Cholesterol (CH) was a gift from the Nile Company for Pharmaceuticals and Chemical Industries (Cairo, Egypt). Streptozotocin (STZ) was obtained from Sigma–Aldrich (St. Louis, MO, USA). Vanadium trichloride, N-1-naphthyl-ethylenediamine, sulfanilamide, sodium nitrate (NaNO_3_), and zinc sulfate (ZnSO_4_) were purchased from Sigma–Aldrich (Munich, Germany). All other chemicals and reagents used were of analytical grade.

### 2.1. Formulation of Bilosomes (LUT-BLs)

Bilosomes (BLs) were developed using the thin-film hydration technique as described by Matloub et al. [28], with slight modification. In brief, the weighed amounts of LP, surfactant (span 60), and CH were solubilized in 10 mL of a chloroform/methanol mixture (2:1) in a round-bottom flask with the aid of an ultrasonic bath sonicator (Ultrasonic bath sonicator, Thomas Scientific, Model SH 150-41; Swedesboro, NJ, USA) for 10 min. Then, in order to obtain a thin, completely dry film, the organic solution was evaporated at 65 °C under reduced pressure for 30 min using a rotary evaporator (Rotavapor, Heidolph VV 2000; Heidolph Instruments, Kehlheim, Germany). Then, the obtained dry film was hydrated with 10 mL of a 0.1 M phosphate-buffered saline (PBS, pH 7.4) hydration solution containing SDC (10 mg% or 25 mg%). Then, the hydrated lipid film was sonicated in a bath sonicator at 40 °C until a milky dispersion formed. Next, the particle size of BL dispersion was decreased using an ultrasonicator (UP100H, Hielscher Ultrasonics GmbH, Teltow, Germany) for 5 min.

### 2.2. Experimental Design

A 2^3^ factorial design was adopted for the design and analysis of the different experimental trials to determine the optimal formulation using Design-Expert1 software (Stat-Ease, Inc., Minneapolis, MN, USA). According to preliminary investigations, the variables of optimization encompassed were surfactant concentration (A), CH:LP molar ratio (B), and SDC concentration (C), which were used at low and high levels. Eight formulae were prepared to cover the effect of the previously mentioned variables on the major quality attributes, including vesicle size (R1, VS), % entrapment efficiency (R2, %EE), zeta potential (R3, ZP), and % drug released after 8 h (R4, %Q8 h), as demonstrated in Table 1. The compositions of different runs are expressed in Table 2.

### 2.3. Characterization of Prepared LUT-BLs

#### 2.3.1. Particle Size (PS), Zeta Potential (ZP, ζ), and Polydispersity Index (PDI)

Bilosome vesicular size was determined using a Zetasizer Nano ZS (Malvern Instruments, Malvern, UK) at 25 °C. Before measurement, 1 mL of the vesicular dispersion was diluted with deionized water. Glass cuvettes were used for PS and PDI measurements, and ζ-cells were used for ZP measurements. All measurements were performed in triplicate and reported as the mean ± SD [29].

#### 2.3.2. Percentage Entrapment Efficiency (%EE) 

The entrapment efficiency (EE) of LUT was measured using ultrafiltration techniques via Centrisart^®^ (MWCO 100000; Sartorius, Marlborough, MA, USA) [26]. Briefly, 2.5 mL of LUT-BLs was placed in the outer compartment and underwent centrifugation at 4000 rpm for 15 min. The free unentrapped LUT was collected from the inner compartment tube after centrifugation, and it was quantified spectrophotometrically at λmax 350 nm (Shimadzu UV spectrophotometer, 2401/PC; Shimadzu Corporation, Kyoto, Japan) [27]. All measurements were performed in triplicate. The %EE was calculated using the following equation:%EE = (Total drug − Free drug)/Total drug × 100 (1)

#### 2.3.3. In Vitro Drug Release

In vitro release studies were performed by the dialysis tubing technique using 50 mL phosphate buffer with pH 6.8 in a water bath shaker (Memmert GmbH, Kupfer, Dominik, Germany) at 100 rpm and 32 ± 2 °C. We then placed 1 mL of luteolin suspension and luteolin bilosome vesicular dispersion equivalent to 10 mg% in a cellulose dialysis bag (Visking^®^ of MWCO 12,000–14,000 Da, Serva, CA, USA) sealed at both ends with Medicell clips (Spectrum). At predetermined time intervals, 2 mL aliquots of the receiver medium were collected and replaced with an equal volume of freshly prepared medium to maintain both a constant release volume and sink conditions. We quantified luteolin spectrophotometrically using a Shimadzu UV spectrophotometer at 350 nm [25]. All samples were estimated in triplicate.

#### 2.3.4. Transmission Electron Microscopy (TEM)

Morphology and the vesicle size of the optimized LUT-BLs were examined by transmission electron microscopy (JEM-1400, JEOL, Akishima, Japan). The sample was properly diluted with double-distilled water and then placed on a 200-mesh carbon-coated copper grid. The sample was stained using a saturated uranyl-acetate-negative stain solution for 30 s before microscopic examination [30].

### 2.4. In Vivo Study

#### 2.4.1. Animals

Swiss albino male mice (18–22 g) were obtained from the animal house of the National Research Center, Cairo, Egypt. At least 1 week of habituation was allocated for the animals before starting the experiment. Four to five mice per cage were housed under humidity- and temperature-controlled conditions with a 12 h light/dark cycle and free access to food and water. This study was approved by the Institutional Animal Care and Use Committee of Cairo University (CU-IACUC) (permit number: CU-III-F-36-20) and complied with the *Guide for the Care and Use of Laboratory Animals* from the US National Institutes of Health (NIH Publication No. 85-23, revised 2011). All efforts were made to minimize animal discomfort and suffering.

#### 2.4.2. Sporadic Alzheimer’s Disease (SAD) Induction

Streptozotocin (STZ, methyl nitrosourea) was injected intracerebroventricularly (ICV) using the method described by Pelleymounter et al. [31] with modifications by Warnock to avoid cerebral vein penetration [30,32,33]. Mice were anesthetized by intra-peritoneal administration of thiopental (5 mg/kg), and we stabilized their head by downward pressure above the ears. Next, a needle was inserted directly through the skin and skull into the lateral ventricle, directed by visualizing an equilateral triangle between the eyes and the center of the skull to locate the bregma. This allowed the needle to be inserted at the following coordinates from the bregma: 1 mm mediolateral, −0.1 mm anteroposterior, and −3 mm dorsoventral. The mice behaved normally nearly 1 min after the injection. 

#### 2.4.3. In Vivo Experimental Design

The animals were separated randomly into four groups (8–10 mice/group). The normal control group got an ICV injection of saline once and for 21 successive days (normal control group). The STZ control group received STZ (3 mg/kg, ICV) once (SAD model group). The third group got STZ (3 mg/kg, ICV), followed by i.n. administration of the LUT suspension (50 mg/kg) after 5 h and then every day for 21 successive days. The fourth group received STZ (3 mg/kg, ICV), then LUT-BLs (equivalent to 50 mg/kg) administered intranasally after 5 h and then every day for 21 consecutive days. A schematic representation of the experimental design is shown in Figure 1.

#### 2.4.4. Intranasal Administration

To administer the formulations intranasally, a single daily dose (5 µL) of the formulation was dropped using a polyethylene tube attached to a micropipette inserted approximately 3 mm into the mouse’s right nostril without anesthesia. After the nasal drop, we kept the animals in a supine position to allow the drug to get into the olfactory region or the upper part of the nasal cavity, where it would have immediate access to the brain [10]. We performed all administrations at similar times of day to prevent deviations in the animals’ performance [4].

#### 2.4.5. Behavioral Assessment

We performed behavioral tests 24 h after the last drug dose. These tests were organized from the least stressful to the most stressful and were conducted under top lighting to reduce the potential for circadian variability.

##### Y-Maze Test

Short-term memory spontaneous alternation behavior was assessed using the Y-maze test. The apparatus consisted of three metallic arms forming a Y-shaped maze. Each arm was 35 cm long, 10 cm wide, and 25 cm high and extended at a 120° angle from the center of the platform. Usually, normal mice prefer to explore a new arm of the maze rather than a familiar one. The test was performed on two consecutive days. On the first (training) day, each mouse was placed at the platform center and allowed to freely move through the maze for 8 min. On the test day, during the 8-min session, the sequence of arms entered by each mouse was recorded. After testing each mouse and between each session, the maze was mopped with ethanol (70%) to remove any olfactory stimulus and decrease the risk of errors in the observations. We considered successive entries into all three arms, known as overlapping triplet sets (real alternation) and the total number of entries into the arms (possible alternations). Then, we calculated the percentage of spontaneous alternation behavior using the following equation [34]:(2)$$\%Spontaneous\;alternation=\frac{Actualal\;ternations}{Total\;number\;of\;possible\;alternations}\times 100$$

##### Morris Water Maze (MWM)

The mice’s visuospatial memory and learning ability was assessed using a MWM test [35]. The apparatus consisted of a large circular stainless-steel pool (150 cm in diameter and 60 cm in height) half-filled with room-temperature water [34]. Two threads placed perpendicular to one another divided the pool into four quadrants. A submerged 10 cm wide, 28 cm high black platform was placed 2 cm below the water surface inside the target quadrant. The platform remained in the same location throughout the test. A nontoxic purple dye was added to make the water opaque and the platform invisible. Normal animals usually learn the platform location and rapidly swim toward it. The test was performed on five consecutive days [36]. On the first 4 days, we subjected each mouse to two consecutive trials with an interval of at least 15 min between each trial. Each trial lasted a maximum of 60 s. When mice located the hidden platform within 60 s, we left them there for 20 extra seconds before removing them. However, when mice could not find the platform, they were gently directed onto it and permitted to stay there for 20 s. Then, the mean escape latency was determined as the time that each mouse took to locate the hidden platform. Consequently, it was measured throughout the four days of the test [37]. On the 5th day, the platform was removed from the pool, and we submitted the mice to a probe-trial session. The mice were allowed to explore the pool for 60 s, and we measured the total time spent in the target quadrant, where the hidden platform used to be [38].

#### 2.4.6. Brain Processing

After assessing cognitive performance, each group was split into two sets, and the animals were sacrificed under light anesthesia via cervical dislocation. Later, the brain was rapidly dissected and washed with ice-cold saline. In the other set, we excised the hippocampal tissues from each brain on an ice-cold glass plate. Next, the hippocampi were homogenized in ice-cold physiological saline (10% *w*/*v*) using a Potter–Elvehjem tissue grinder (Thermo Scientific, Flemington, NJ, USA) at a speed of 14,000 rpm for 20 s to quantify various biochemical parameters. Finally, the supernatants were stored at −80 °C in an Isotemp freezer (Basic Thermo Fisher Scientific, Carlsbad, CA, USA) until analysis [31].

#### 2.4.7. Measurement of Biochemical Parameters

##### Estimation of Proinflammatory Mediators (TNF-α)

We quantified hippocampal TNF-α using ELISA kits obtained from RayBiotech Inc. (Norcross, GA, USA). We were able to perform all the procedures following the manufacturer’s guidelines. The results are reported in pg/g tissue for TNF-α [35].

##### Estimation of Amyloidogenesis and Tauopathy (Aβ1-42 and Tau)

We quantified hippocampal Aβ1-42 and tau using mouse ELISA kits obtained from Novus. We performed all the procedures following the manufacturer’s guidelines. The results are in ng/g tissue for tau and pg/g tissue for Aβ1-42 [35,39].

##### Determination of Mouse MMP-9 (Matrix Metalloproteinase 9)

We measured hippocampal MMP-9 using ELISA kits (Wuhan Fine Biotech Co., Ltd., Wuhan, China). We performed all the procedures following the manufacturer’s guidelines. The results are in ng/g tissue.

#### 2.4.8. Histopathology Study

The brain samples were flushed and fixed in 10% neutral buffered formalin for 72 h. Then, samples were processed in different grades of ethanol, cleared with xylene, and infiltrated with Paraplast Plus tissue-embedding media (Leica Biosystems, Germany). Using a rotatory microtome, we cut 4 µm thick serial sagittal brain sections and mounted them on glass slides to examine the hippocampal subregions. We then stained the tissue sections with hematoxylin and eosin for pathological examination. We stained another set of tissue sections with toluidine blue to quantify intact neurons and examined them using an optical microscope. We used standard sample-fixation and staining procedures and protocols described by Culling [40]

#### 2.4.9. Immunohistochemical Staining and Analysis

Immunohistochemical staining of GFAP, IbA-1, and Aβ was conducted according to the manufacturer’s protocols and directions. First, antigen-retrieved brain sections were blocked with 3% hydrogen peroxide in methanol for 15 min and then incubated overnight at 4 °C with the primary antibodies (antiglial fibrillary acidic protein as a monoclonal antibody (sc-166458, Santa Cruz Biotechnology, Inc., Dallas, TX, USA, dilution of 1:200), anti-Iba1 antibody (ab108539-Abcam, 1:100), and anti-Aβ antibody (ab201060-Abcam, 1:500)). Next, the sections were washed with PBS three times and incubated with a secondary antibody HRP Envision kit (DAKO) for 20 min. Then, they were washed with PBS three times, incubated with diaminobenzidine for 10 min, counter-stained with Mayer’s hematoxylin, then dehydrated and cleared in xylene. Finally, samples were cover-slipped for microscopic examination.

##### Quantitative Immunohistochemical Analysis

We analyzed six random non-overlapping fields to determine the positive mean area percentage of immunohistochemical expression levels of GFAP and Aβ in each immune-stained tissue section. We also counted the IbA-1/++ microglial cells in six random non-overlapping fields in each immune-stained tissue section. We performed all morphological examinations, imaging, and quantitative analysis using Leica Application system modules for histological analysis (Leica Microsystems GmbH, Wetzlar, Germany). 

#### 2.4.10. Statistical Analysis

The data were expressed as the mean of triplicate experiments ± standard deviation (SD). Data were analyzed using one-way analysis of variance (ANOVA), followed by the least significant difference procedure using Prism^®^ software (GraphPad Prism software version 8.01, Inc., San Diego, CA, USA). We considered *p* < 0.05 as significant.

## 3. Results and Discussion

### 3.1. Preparation and Characterization of Bilosomes

Bilosomes are modified nanovesicles that were first developed in 2004 via the integration of phospholipids with bile salts. The latter are used as absorption enhancers that enhance the solubility of hydrophobic moieties and increase the fluidity of biological membranes [13,41]. Consequently, such integration yields more robust, nano-sized vesicles with higher negative zeta potential compared to unmodified liposomes [12,42]. The rationale for selecting SDC as a bile salt was based on its previously reported high permeation-enhancing ability with no toxicity [9,11,12,36]. Besides, its high hydrophobic nature (hydrophilic–lipophilic balance, HLB = 16) increases the loading of hydrophobic-active drugs in the phospholipid bilayer [9].

In the present study, different BL formulations were prepared using the thin-film hydration technique based on 2^3^ full factorial design to study the influence of formulation-independent variables on the characteristics of the nanovesicles. These variables include (A) surfactant concentration, (B) LP: cholesterol ratio, and (C) sodium deoxycholate concentration, and were studied through eight experimental runs.

### 3.2. Experimental Factorial Design

Results in Table 3 show that the predicted R2 values for different investigated responses were in reasonable agreement with the adjusted R2. For all responses, low coefficient of variation was observed in the obtained quadratic model, confirming that the model can be used to explore the design space. The model yielded different polynomial equations for responses R1, R2, R3, and R4 (Equations (3)–(6)). The effect of the variables on responses was explained from the polynomial equations’ signs (+ or −), indicating the agonistic or antagonistic effects of the variable on the response. An ANOVA test for the observed data of R1 (VS), R2 (EE%), R3 (ZP), and R4 (%Q8 h) proved that the model was significant and the data were fitted. The yielded a non-linear model for different responses exhibited a significant *p* value < 0.0001. As shown in Table 3, F value was 1035.8, 637.00, 93.91, and 235.88, and the *p* value was found to be 0.0238, 0.0303, 0.0478 and 0.0398, which implies the model is significant. Figure 2 shows Pareto charts for the effect of different variables on responses. 

#### 3.2.1. Effect of Independent Variables on Vesicle Size (R1, VS)

Vesicle size can significantly affect the drug penetration rate, as when it decreases, it can penetrate deeper within the tissue compared to the larger vesicles. Therefore, the VS of prepared LUT-BLs was investigated to understand the high penetration offered by BLs. Our results shown in Table 2 revealed that the vesicle size of all BLs ranged from 153.2 ± 0.98 nm to 250.0 ± 2.14 nm. It was reported that efficient intranasal brain targeting occurs when nanocarrier particles are smaller than 250 nm [4]; therefore, the prepared BLs were in good agreement with the study aim. The effect of different variables is shown in contour plots and 3D plots (Figure 3 and Figure 4). The obtained polynomial equation Equation (3) is as follows:(3)$$R_{1}=+207.13-17.38*A+14.13*B-17.38*C-4.37*A*B-5.87*A\;*C+4.13*B*C$$

ANOVA testing showed that surfactant concentration (A) had a negative significant effect (*p* = 0.0311) on VS; by increasing the surfactant concentration, the VS was significantly decreased. This finding might be due to the fact that surfactants can efficiently decrease the interfacial tension between the water, LP, and CH, thereby reducing the distance between them, causing a decrease in vesicle size. Data revealed that changing CH content in higher CH:LP ratio (B), had a synergistic effect (*p* = 0.0137) on the vesicle size of BLs. The increase in CH concentration restrained the vesicles’ lipid close packing, which resulted in a higher distribution of aqueous phase within the vesicle, thereby obtaining vesicles with increased size. Furthermore, increasing the amount of CH can cause a significant increase (*p* = 0.0169) in LUT entrapment within the vesicles’ bilayer, and can therefore increase the size of the vesicles [26]. As depicted in Figure 3a–c and Figure 4a–c, an effective decrease in VS was also observed by increasing SDC (C) due to the enhanced flexibility of BLs. A significant decrease in the VS of the obtained BLs was observed by increasing SDC concentration (Figure 3a–c and Figure 4a–c). This result can be attributed to the enhanced flexibility of bilosomes [43].

#### 3.2.2. Effect of Independent Variables on Entrapment Efficiency Percent (R2, EE%)

The %EE depends on the type and amount of bile salts and the nature of the incorporated drug. In the current study, loading LUT in BLs yielded a sufficient EE% ranging from 70.4 ± 0.77% to 95.1 ± 0.47% (Table 2). The hydrophobic nature of SDC (HLB =16) in the BL structure may explain this. Hydrophobicity helped in the intermolecular integration of the hydrophobic LUT within the hydrophobic part of the phospholipid bilayer, yielding a high %EE%. El-Nabarawi et al. reported concurrent findings [13], with a higher %EE of dapsone in BLs prepared with SDC rather than other types of bile salts with higher HLB values (lower hydrophobicity). The effect of independent variables on the response was represented by contour plot and 3D plots (Figure 3d–f and Figure 4d–f). The data plot and obtained polynomial equation (Equation (4)) show that all the independent factors had a significant effect on EE%.
(4)$$R_{2}=+82.63-6.62\;*A+3.38\;*B-0.37*C+1.12*A*B-1.63*A*C-\;0.63*B*C$$

Figure 3d–f and Figure 4d–f show that the increase in the concentration of surfactant (A) was found to adversely affect the %EE (*p* = 0.0121). The observed decrease in EE% may be related to the phenomenon of partition. High content of surfactants will increases the partition of the drug with increasing drug solubility in dispersion medium due to the possible existence of mixed micelles, thereby decreasing the EE% [44]. It was observed that CH:LP ratio (B) had a positive effect on EE%, and an increasing concentration of CH was accompanied by a high EE%. Cholesterol can increase the hydrophobicity and rigidity of the lipid bilayer membrane, which results in improved stability and increased the ability to entrapment drugs within bilosomal vesicles [45]. Additionally, a negative sign obtained by the polynomial equation (Equation (4)) showed that SDC concentration (C) had a significant influence on EE%. An increase in SDC concentration from 10 mg% to 25 mg% was accompanied by a decrease in LUT EE%. This may be because increasing the bile salt (SDC) concentration can cause the formation of mixed micelles, thus increasing the solubility of hydrophobic drugs and lowering the EE% [9,11,46]. Furthermore, at a higher SDC concentration (25 mg%), the lipid membrane is more fluid, allowing the entrapped drug to leak out [9]. Mosallam et al. reported comparable results; they observed a lower EE% for the hydrophobic terconazole by increasing the bile salt amount [15]. Additionally, El-Nabarawi MA et al. [13] justified the decreased EE% of the hydrophobic dapsone upon increasing SDC concentration by the higher drug solubilization, which is in agreement with our findings.

#### 3.2.3. Effect of Independent Variables on Zeta Potential (R3, ZP)

ZP is considered an important measure of the net charges attained by vesicular systems and is attributed to the stability of prepared colloidal dispersions. Nanovesicles with ZP values around ±30 mV are considered stable and cause an electrostatic repulsion between charged particles, which prevents particle agglomeration, resulting in an increase in nanovesicle dispersion stability [47]. Our results show that all the prepared bilosomes exhibited high zeta potential values ranging from −31.0 ± 0.88 mV to −42.8 ± 0.24 mV (Table 2), which reflect good physical stability and low aggregation tendency. The negative sign for ZP could be attributed to the presence of anionic SDC in the vesicular structure at pH 7.4 [9]. In addition, LUT has reported pKa values of 6.9, 8.6, and 10.3; therefore, at a pH of 7.4, it would be in a monoanionic form, which further increases the negative ZP values [17,23,48]. The effect of the surfactant concentration (A), CH:LP ratio (B), and bile salt concentration (C) on ZP of LUT-BLs is graphically illustrated in 3D surface and contour plots (Figure 3g–i and Figure 4g–i). The polynomial equation (Equation (5)) for the model is as follows:(5)$$R_{3}=+36.69+1.19*A-0.26*B+4.26*C+0.24*A*B+0.26*A*C-\;0.19*B*C$$

ANOVA testing revealed that both surfactant concentration (A) and CH:LP ratio (B) were non-significant with *p*-values of 0.0997 and 0.3949, respectively. Only changing bile salt concentration (C) had a synergistic effect (*p* = 0.0280) on ZP. Results show that increasing the concentration of the anionic SDC can lead to a significant increase in ZP value for prepared BLs. The integration of SDC within the phospholipid bilayer might decrease the surface tension because of its higher negativity, which causes a repulsion between the vesicles. These findings were consistent with previously reported data by Abdelalim et al. [49] and Elnaggar et al. [12].

#### 3.2.4. Effect of Independent Variables on % Drug Released after 8 h (R4, %Q8 h)

The in vitro release profiles depicted in Figure 5 showing the cumulative amount of released LUT from LUT suspension and the prepared LUT-loaded BLs as a function of time. The %Q at 0.5 h of LUT from LUT suspension was found to be 2.2 ± 1.68%, which can be attributed to the poor solubility of LUT. The release profiles showed a gradual sustained LUT release from different formulations over 24 h; this might be attributed to the ability of BLs to act as a drug reservoir [16]. Furthermore, the drug-release profiles of LUT from BLs exhibited a biphasic pattern with a rapid initial burst release in the first 30 min (%Q0.5 h, 13.08 ± 2.53% to 29.83 ± 3.45%), followed by a slower release phase. This initial burst might be attributed to the detachment of drug adsorbed on the vesicles external surface; then, the extended phase is attributed to the gradual partitioning of LUT entrapped within the vesicular bilayer to the release medium [50]. The biphasic release profiles of the LUT from all the prepared BLs are expected to offer a great benefit. As an anti-Alzheimer’s drug, the initial released flush, followed by the slower subsequent release for 24 h would keep the patient under treatment throughout the day with decreasing frequency of administration. As Figure 5 illustrates, %Q at 8 h obtained from drug suspension was 7.24 ± 0.81%, whereas it ranged from 42.4 ± 2.00% to 80.0 ± 1.10% for LUT-loaded BLs. We found % Q at 24 h to be 9.75 ± 1.01% for pure drug suspension, whereas it ranged from 92.8 ± 2.12% to 98.7 ± 4.20% for LUT-loaded BLs. A large difference was observed between the results obtained from LUT-loaded BLs and LUT suspension; the release rate of LUT from LUT BLs was highly enhanced compared with that from LUT suspension. The improvement in drug dissolution may come from the enhanced LUT solubility due to presence bile salts in parallel with the reduced size of the vesicles, leading to a nano-solubilization effect [13,51]. It was observed that BL2 had the lowest % LUT released (%Q at 0.5 h, 13.08 ± 2.53; %Q at 8 h, 42.4 ± 2); this might be attributed to vesicle size because BL2 had the largest vesicle size and therefore had the most reduced values of surface area exposed to the release medium and eventually obtained the slowest rate of release [13].

Contour plots and 3D plots (Figure 3j–l and Figure 4j–l) were generated to investigate the effect of different independent variables on different responses. The model showed the following polynomial equation describing the model:(6)$$R_{4}=+63.36+8.91*A-5.91*B+6.16\;*C-0.36*A*B-2.89*A*C-\;0.61*B*C$$

Both surfactant concentration (A) and SDC concentration (C) were found to exhibit a positive significant effect (*p* = 0.0248 and 0.0348, respectively) on %Q at 8 h of LUT from BLs. At their higher concentration, they can increase the flexibility of the prepared vesicles and thereby increase the percent LUT released from BL vesicles. Obviously, as CH content increased by changing the CH:LP ratio (B), %Q at 8 h was gradually decreased. These results can be explained by the fact that with increasing CH concentration, the BL vesicle wall becomes stiffer, which can restrain the drug release rate. Furthermore, CH can retard the permeability or liberation of the drug entrapped within the prepared vesicles by diminishing the fluidity of the vesicular membrane [48]. 

#### 3.2.5. Selection of Optimized LUT-BLs

The aim of the optimization process was to determine the best formula with the optimal levels of dependent variables in order to prepare a pharmaceutical product with high quality to achieve the aim of the study. In this study, numerical optimization was conducted using Design-Expert^®^ software, depending on the desirability function to overcome the multiple and opposing responses. The suggested optimized LUT BL (BL 4) showed the highest desirability of 0.898. The optimized LUT BL (BL4) that was found to fulfil the optimization criteria (minimum values for VS, maximum values of EE%, ZP and %Q8 h) was prepared with 100 mg surfactant (span 60), a CH:LP ratio of 2, and 25 mg SDC. This optimized formulation showed VS of 153.2 ± 0.98 nm, %EE of 70.4 ± 0.77%, ZP of −42.8 ± 0.24 mV, and %Q8 h of 80.0 ± 1.10%. Table 4 showed the predicted and observed dependent variables of the optimized BL4 to reinforce the optimization process validity. Obviously, a high similarity between the observed and predicted values was depicted; therefore, BL4 was selected for further investigation. 

### 3.3. TEM of the Optimized BLs

The morphology of the optimized formula (BL4) was assessed by TEM, as shown in Figure 6. The TEM micrograph showed homogenous, unilamellar nanovesicles without aggregation. They were well separated and spherical without micelles. The obtained regular spherical vesicles may be attributed to the increased flexibility and the decrease in surface tension [52]. It is worth mentioning that the size obtained by TEM is smaller than that obtained from the dynamic light scattering technique using a Zetasizer NanoZS (Malvern Instrument) because the principle involved in the analysis in both techniques is different. Regarding dynamic light scattering (DLS), it is an intensity-based technique whereby the resultant size distribution is the average hydrodynamic size of the nanoparticles and is often affected by the existence of large particles, dust, or aggregates [53]. In particular, the nanoparticles measured using DLS techniques are in solution surrounded by stagnant layers of the used medium, which enlarge the measured diameter of the nanoparticles. On the other hand, microscopic analysis by TEM is mainly based on nanoparticle tracking analysis (NTA), and observation is generally performed after the routine procedure of air drying of a nanoparticle-containing droplet on the TEM grid. NTA is a number-based technique that tracks individual nanoparticles (single-particle tracking) [53]. Therefore, the latter can provide an accurate number-based average dimension with minimum bias for samples free from artifacts [54]. Therefore, DLS will obtain a large size than TEM analysis. Although the two techniques showed different vesicles size, BL4 showed a good homogeneity reflected from the PDI value = 0.219 ± 0.012, which indicated the narrow size distribution; therefore, more stable and homogenous formulations were obtained.

### 3.4. In vivo Study

Streptozotocin (STZ) was first established as an anti-cancer agent. Its systemic administration was found to cause type 1 diabetes in animal models [55]. However, after ICV administration at a sub-diabetogenic dose, it reduces glucose uptake in the brain and activates pathological and neurobehavioral features resembling AD [34,55,56]. The deficiency of brain insulin signaling initiated by ICV-STZ administration leads to neurodegeneration; thus, we used the ICV-STZ mouse model as an AD animal model.

#### 3.4.1. Behavioral Assessment

##### Y-Maze Test

The Y-maze evaluates the willingness of rodents to discover new environments [57]. The STZ control group had a significantly lower (about twofold) percentage alternation than the normal control group. These results confirmed that STZ administration reduced spatial and working memory, as stated by Fronza et al. [58]. The LUT-SUSP and LUT-BS groups both displayed better short-term memory acquisition by approximately 57% and 65%, respectively, than the STZ control group. These results are consistent with previously reported data that revealed improvements in spatial recognition memory in the Y-maze test after luteolin administration [59]. Furthermore, the LUT-BLs group had significantly better test results than the LUT-SUSP group, as shown in Figure 7A. These results reveal that BLs enhanced LUT activity. This useful effect may result from the nanosolubilization of the poorly soluble LUT, which, in turn, enhances drug dissolution and absorption [40].

##### Morris Water Maze (MWM)

The Morris water maze is a behavioral test assessing hippocampal-dependent spatial learning and long-term spatial memory in mice [4,45]. The mouse’s swimming distance before locating the platform is identified as the most consistent indicator of memory and learning [46]. We recorded the mean of two trials completed for each group on each day. On the first day, all mice displayed comparable mean escape latency values. In the following days, the LUT-SUSP group had a significantly lower mean escape latency than the STZ control group. We recorded differences of 26, 23, and 17 s on the 2nd, 3rd, and 4th days, respectively. These data coincide with a those reported in previous study that showed that luteolin-fed mice could use the visual cues of the extra maze to resolve the acquisition task, demonstrating enhanced spatial learning and memory capability [46]. The LUT-BS group showed significantly better results than the LUT-SUSP group, with no significant differences in mean escape latency relative to the normal control group throughout the testing period (Figure 7B). These results highlight the importance of bilosomes in terms of enhancing luteolin solubility, absorption, brain uptake, and advanced learning and memory function. In conclusion, luteolin bilosomes may meaningfully alleviate cognition deficits and could be more effective at much lower doses than the advised therapeutic dose.

##### Morris Water Maze and Time Spent in the Target Quadrant

On average, the normal control group spent twice as much time in the target quadrant as the STZ control group (*p* < 0.05). The LUT-SUSP group also stayed longer in the target quadrant (by 1.6-fold) than the STZ control group (*p* < 0.005) but less (by 1.3-fold) than the normal control group. Furthermore, a non-significant difference was noticed between the LUT-BS group and the normal control group, whereas the LUT-BS group spent 30 s more than the STZ control group in the target quadrant, as shown in Figure 7C. Finally, the LUT-BS group stayed significantly longer in the target quadrant (1.3-fold) than the LUT-SUSP group. 

#### 3.4.2. Biochemical Parameters

##### Quantification of Proinflammatory Mediators (TNF-α)

The STZ control group had significantly higher hippocampal TNF-α levels than the normal control group, which is consistent with previous reports showing that STZ administration increased neuroinflammatory cytokine release in mouse hippocampi [55,60]. As shown in Figure 8A, the luteolin suspension significantly decreased hippocampal TNF-α levels. This result confirms the anti-inflammatory activity of luteolin. Moreover, administering luteolin as bilosomes enhanced this effect. Our findings were coherent with several previous reports regarding the capability of luteolin to decrease the production of inflammatory mediators by lipopolysaccharide-stimulated microglial cells [1,61,62].

##### Quantification of Aβ1-42 and Tau

Many reports suggest that Aβ is a major pathological characteristic of AD [55]. Additionally, tau protein is hyperphosphorylated in AD and accumulates in neurons. This causes abnormal mitochondrial dynamics, which, in turn, decrease the dendritic protein and dendritic spines, resulting in hippocampal-based learning and memory impairments [63]. We observed an elevation in hippocampal Aβ1-42 and tau levels in the STZ control group compared with the normal control group (Figure 8B,C). This is because STZ reduces cerebral glucose uptake, dulls brain insulin receptors, reduces PI3K-AKT signaling activity, and raises the activity of glycogen synthase kinase 3β [64]. These changes induce tau hyperphosphorylation. Furthermore, glucose hypometabolism initiates the process that eventually ends in Aβ aggregation [65]. Conversely, the Aβ1-42 and tau levels were found to be lower in the LUT-SUSP group than in the STZ control group and even lower in the LUT-BS group. These results confirm the ability of luteolin to lower Aβ and induce tau disaggregation [66]. Thus, luteolin bilosomes successfully reduced the pathological changes of AD.

##### Quantification of MMP9

MMP9 has emerged as a key factor in many brain pathologies involving neuroinflammation and neurodegeneration [67]. Elevated hippocampal MMP-9 expression has been associated with the development of cognitive impairment caused by Aβ [65]. The underlying mechanism is that MMP9 induces basal lamina breakdown and gap-junction destruction in the neurovascular unit, enhancing central nervous system permeability and inflammation in AD. Figure 8D shows that the STZ control group had significantly higher MMP9 levels than the normal control group. Moreover, the LUT suspension significantly reduced MMP9 levels, showing that luteolin downregulates MMP9 expression. These results are consistent with the findings of Ali and Siddique [66], who described a decreased permeability and infiltration of leukocytes and other inflammatory agents into the brain after LUT administration. Finally, the LUT-BS group’s MMP9 levels were even lower than those of the LUT-SUSP group, with insignificant difference from those of the normal control group. The obtained results demonstrate the key role that nanoplatforms play in enhancing the therapeutic effectiveness of administered flavonoids. 

#### 3.4.3. Histopathology Study

Microscopic examination of the hippocampal subregion in normal control samples displayed typical morphological highlights of hippocampal CA3, as well as the dentate gyrus zone. CA3 subregion exhibited apparently intact cellular elements, including 5–6 cell thick pyramidal neurons of the middle layer with evident subcellular details (Figure 9A, black arrows). The mean intact pyramidal neuron count in toluidine-blue-stained sections was up to 66 cells/field (Figure 9E). Normally distributed glial cells with normal vasculatures, as well as intact neuropil, were recorded. On the other hand, CA3 subregion of AD-induced model samples revealed abundant records of neuronal loss and damage with many figures of darkly stained, pyknotic, hypereosinophilic and shrunken neurons without obvious subcellular details (Figure 9B, red arrows) paired with moderate intercellular edema and higher records of microglial cells and astrocytic infiltrates (Figure 9B, arrowhead). The mean intact neuron count was up to 13 cells/field in toluidine-blue-stained tissue sections (Figure 9F), which means that histopathological alterations of the brain caused by STZ were like those generated by AD [68].

Moderate neuroprotective efficacy was observed in LUT suspension-treated samples with moderately higher records of intact neurons (Figure 9C, black arrows) alternating with minor focal areas of persistent neuronal degenerative change records (Figure 9C, red arrows). The mean count of intact neurons was up to 49 cells/field in toluidine-blue-stained tissue sections (Figure 9G). Mild persistence records of glial cell infiltrates were shown. Moreover, LUT-BL-treated samples showed almost intact, well organized histological features of CA3 regions, with abundant records of apparent intact neurons and few sporadic records of neuronal degenerative changes (Figure 9D, black arrows). Minimal records of abnormal glial cell infiltrates or edema were shown, with a mean intact cell count was up to 65 cells/field in toluidine-blue-stained tissue sections (Figure 9H). This may be ascribed to the fact that the administration of STZ resulted in histopathological alterations of the brain in the same manner as those generated by AD [68]. In addition, histological lesions detected in the hippocampus clarify the main role of AD in changing memory, as well as spatial learning potentials [69].

Well organized morphological structures were demonstrated in the dentate gyrus subregion, with an apparent intact granule cell layer, as well as an inner hilar region with intact hilar cells demonstrating apparent intact subcellular details (Figure 10A, black arrows). The mean count of intact pyramidal neurons was up to 198 cells/field (Figure 10E). However, dentate gyrus region of SAD-model-induced samples showed moderate records of neuronal degenerative changes of inner small granule cell layers with nuclear pyknosis (Figure 10B, red arrow), with a mean count of intact granule cells up to 168 cells/field (Figure 10F). Evident neuroprotective efficacy was shown in LUT suspension and LUT-BL group dentate gyrus subregions in different samples (Figure 10C,D), with a count of mean intact cells up to 194 and 196 cells/field, respectively (Figure 10G,H).

#### 3.4.4. Immunohistochemical Analysis

We observed a significant increase in GFAP immunohistochemical expression in AD model samples, with mean expression levels up to 6.2 folds compared with mean normal control immunohistochemical area percentage of expression (2.88%). We also observed a significant reduction in the mean area percentage of expression values up to 41.4% in LUT-suspension-treated samples compared with the mean SAD-model-induced group samples. Moreover, higher ameliorative efficacy was recorded in LUT-BL samples, with up to 74% reduction in mean expression levels in comparison with model-induced samples (Figure 11).

The histopathological findings revealed the neuronal degeneration and the formation of Aβ plaques in the hippocampus, which was consistent with the findings of Khalil et al. [69] in an AD model. GFAP is considered an intermediate filament that is mainly expressed in ependymal cells and astrocytes [70]. GFAP affects mitotic activity, astrocyte–neuron interaction, communication between cells, and repair of CNS injury [71]. Prominent astrogliosis demonstrated in GFAP, immune staining of brain sections mainly around the amyloid plaques confirmed the role of astrocytes in the degradation of amyloid plaques via the astrocytic processes [72]. We observed a significant increase in the mean cell count of activated IbA1/++ microglial cells of up to sixfold elevation in model-induced samples in comparison with normal control samples. However, a significant decrease was observed in activated IbA1/++ microglial cells count in LUT suspension and LUT-BL-treated samples of up to 34% and 71%, respectively, in comparison with model-induced samples (Figure 12). Quantitative analysis of intraneuronal/++ beta amyloid mean expression levels exhibited up to 6.4% of induced model samples of pyramidal neurons with positive immune expressions. However, a significant reduction was observed in LUT suspension samples of up to 0.95% of pyramidal neurons. Moreover, minimal expression levels were shown in LUT-BLs and normal control samples, with up to 0.08% and 0.05%, respectively (Figure 13).

## 4. Conclusions

To our knowledge, this study was the first evaluation of the antioxidant, anti-inflammatory, and antiamyloidogenic potentials of luteolin-loaded bilosomes. Such novel bilosomes take advantage of non-invasive intranasal delivery and are more efficient than the corresponding suspension form, allowing for lowering of the administered dose. Thanks to histopathological and biochemical assessments, we were able to demonstrate the multifaceted effects of luteolin-loaded bilosomes. We also showed that they notably improved short- and long-term spatial memory through neurobehavioral assays. In short, our preclinical study on animals demonstrated a successful development of effective, non-invasive, and safe LUT-loaded nanobilosomes as a promising therapeutic tactic to manage SAD. Therefore, our promising results pave the way for a deep dive into the future of clinical trials.

## Figures and Tables

**Figure 1 pharmaceutics-14-00576-f001:**
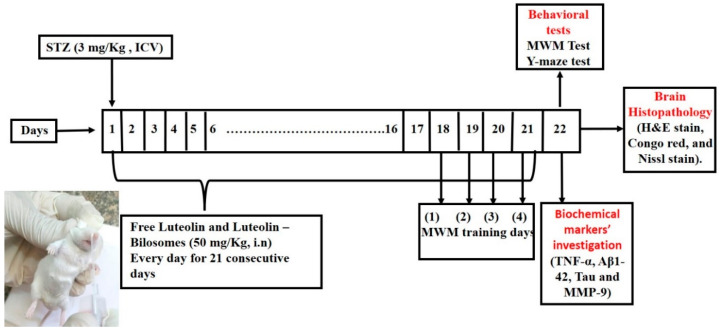
A schematic representation of the in vivo experimental design.

**Figure 2 pharmaceutics-14-00576-f002:**
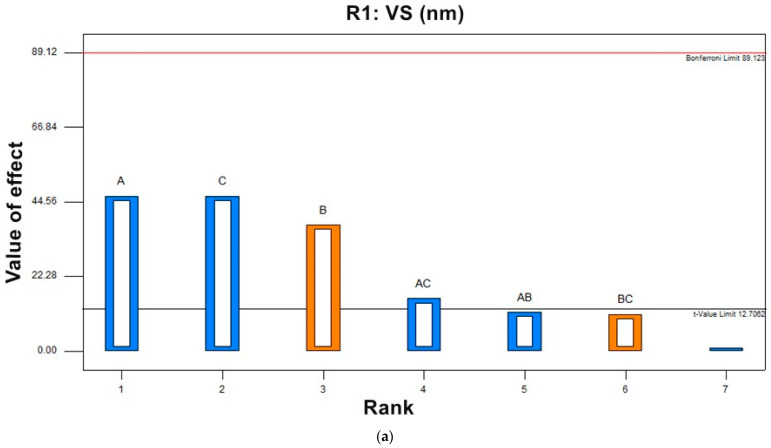
Pareto charts of the effect of independent variable (A) surfactant concentration, (B) CH:LP molar ratio, and (C) SDC concentration each variable alone or combined (AC), (AB) and (BC) on (**a**) R1 (VS), (**b**) R2 (EE%), (**c**) R3 (ZP), and (**d**) R4 (%Q8 h).

**Figure 3 pharmaceutics-14-00576-f003:**
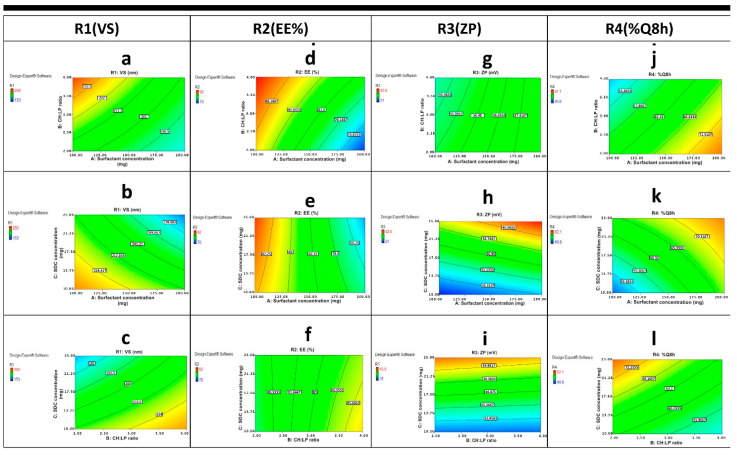
Contour plots of the effect of variables; (A) surfactant concentration, (B) CH:LP molar ratio, and (C) SDC concentration on different of responses R1 (VS), R2 (EE%), R3 (ZP) and R4 (%Q8 h) following: (**a**) the effect of A and B on R1, (**b**) the effect of A and C on R1, (**c**) the effect of B and C on R1, (**d**) the effect of A and B on R2, (**e**) the effect of A and C on R2, (**f**) the effect of B and C on R2, (**g**) the effect of A and B on R3, (**h**) the effect of A and C on R3, (**i**) the effect of B and C on R3, (**j**) the effect of A and B on R4, (**k**) the effect of A and C on R4, and (**l**) the effect of B and C on R4.

**Figure 4 pharmaceutics-14-00576-f004:**
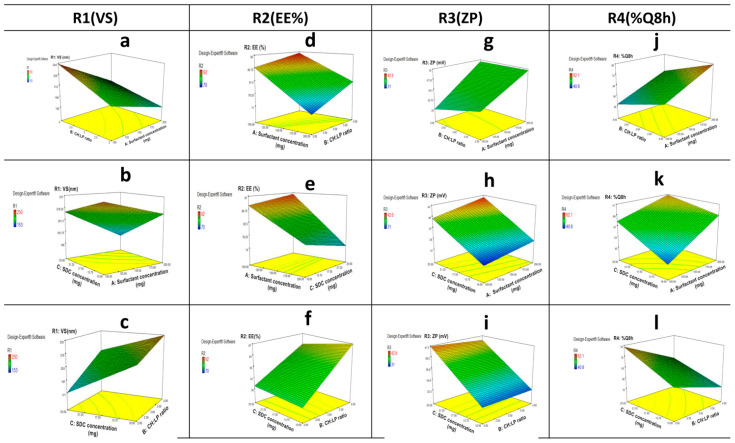
3D surface plots of the effect of variables; (A) surfactant concentration, (B) CH:LP molar ratio, and (C) SDC concentration on different responses following: (**a**) the effect of A and B on R1, (**b**) the effect of A and C on R1, (**c**) the effect of B and C on R1, (**d**) the effect of A and B on R2, (**e**) the effect of A and C on R2, (**f**) the effect of B and C on R2, (**g**) the effect of A and B on R3, (**h**) the effect of A and C on R3, (**i**) the effect of B and C on R3, (**j**) the effect of A and B on R4, (**k**) the effect of A and C on R4, and (**l**) the effect of B and C on R4.

**Figure 5 pharmaceutics-14-00576-f005:**
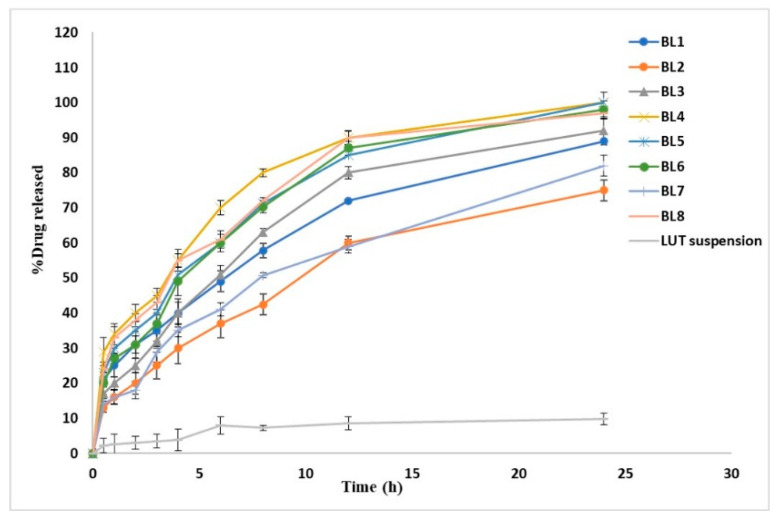
In vitro release profile of luteolin in phosphate buffer (pH 6.8) from LUT suspension and different LUT-loaded BLs. Results are represented as mean ± SD, *n* = 3.

**Figure 6 pharmaceutics-14-00576-f006:**
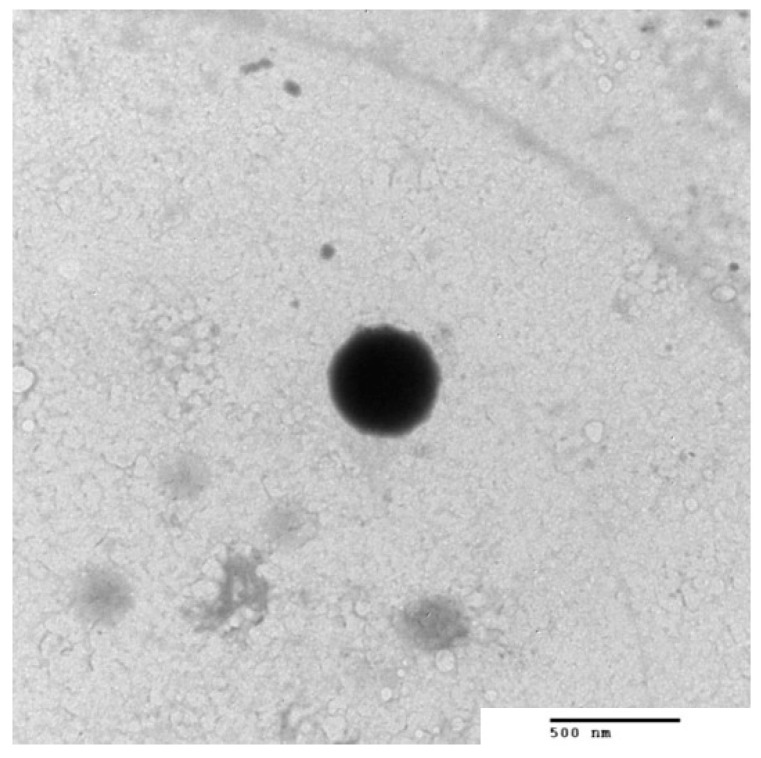
TEM of luteolin-loaded bilosomes (BL4) (150,000× magnification).

**Figure 7 pharmaceutics-14-00576-f007:**
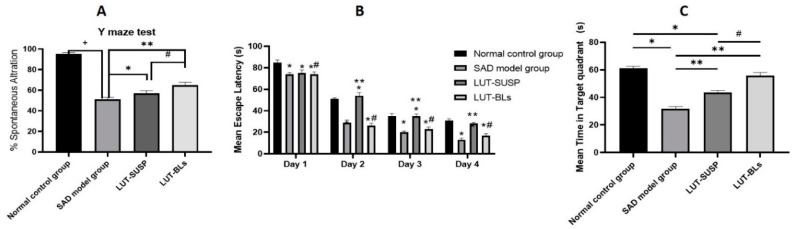
Behavioral assessment of the effect of luteolin bilosomes on (**A**) the percentage of spontaneous alternation, (**B**) mean escape latency in MWM, and (**C**) time spent in the target quadrant in MWM. Animals were divided into four groups (n = 8 in each group), one of which was designated as the normal control group. The second group received STZ (3 mg/kg, ICV) and was thus denoted the STZ group. The remaining two groups were all injected first with STZ (3 mg/kg, ICV) followed by intranasal administration of luteolin suspension (50 mg/kg every day for 21 days) and luteolin-loaded bilosomes (50 mg/kg i.n. for 21 days), respectively. Statistical analyses were performed using one-way analysis of variance (ANOVA) followed by the Tukey post hoc test. +: Statistically significant different from the normal control group (saline) at *p* < 0.05; * and **: statistically significant different from the positive control group (STZ, 3 mg/kg) at *p* < 0.05; #: statistically significant different from the luteolin suspension group (50 mg/kg) at *p* < 0.05. Luteolin bilosomes (equivalent to 50 mg/kg).

**Figure 8 pharmaceutics-14-00576-f008:**
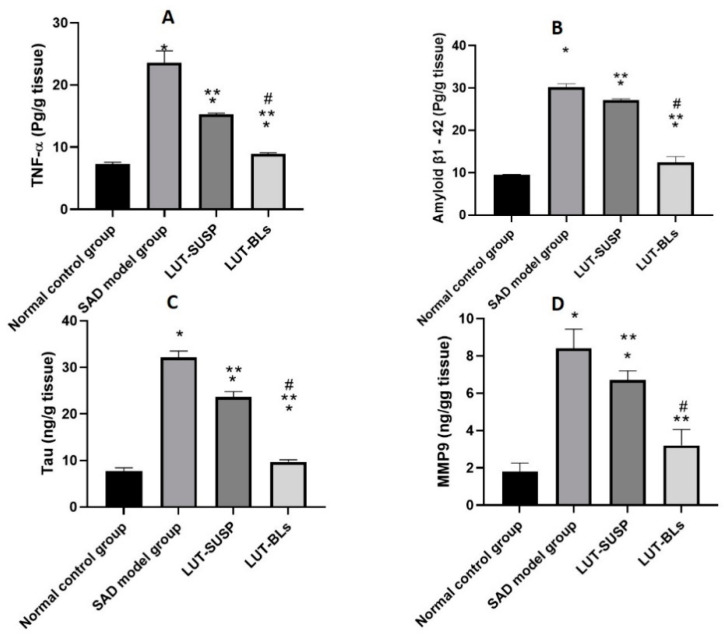
Biochemical analysis of (**A**) TNF-α level, (**B**) amyloid β1-42 level, (**C**) tau level, and (**D**) MMP9 level. Animals were divided into four groups, one of which was designated as the normal control group. The second group received STZ (3 mg/kg, ICV). The remaining two groups were injected first with STZ (3 mg/kg, ICV), followed by injection with luteolin suspension (50 mg/kg, i.n. for 21 days) and luteolin-loaded bilosomes (50 mg/kg, i.n. for 21 days), respectively. The brain of the animals (n = 8) in each group was retrieved and homogenized as 10% homogenate in 0.1 M PBS before being centrifuged, and supernatants were used further analyses. Statistical analyses were performed using one-way analysis of variance (ANOVA) followed by the Tukey–Kramer post hoc test, whereby each value was presented as mean ± standard deviation (SD). * Statistically significantly different from the normal control group (*p* < 0.05); ** statistically significantly different from the STZ group, 3 mg/kg (*p* < 0.05); # statistically significantly different from the LUT suspension group, 50 mg/kg (*p* < 0.05).

**Figure 9 pharmaceutics-14-00576-f009:**
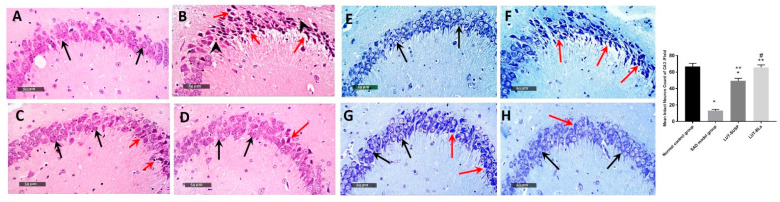
Microscopic examination of the neuroprotective histopathological effect of different groups on CA3 /hippocampal subregions stained with H&E stain (**A**) normal controls, (**B**) model AD samples, (**C**) LUT suspension samples, and (**D**) LUT-BLs samples. Microscopic examination of toluidine-blue-stained pyramidal neurons in CA3/hippocampal subregions in different groups. (**E**) normal controls, (**F**) model AD samples, (**G**) LUT suspension samples, and (**H**) LUT-BLs samples. Black arrows = intact neurons; red arrows = damaged neurons. Data are expressed as means ± SD. * Statistically significant difference from normal control group; ** statistically significant difference from the SAD model group; and # statistically significant difference from LUT suspension group.

**Figure 10 pharmaceutics-14-00576-f010:**
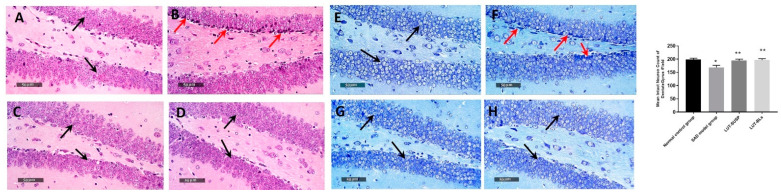
Neuroprotective histopathological effect of different treatments on dentate gyrus/hippocampal subregions of AD model stained with H&E stain. (**A**) Normal controls, (**B**) model AD samples, (**C**) LUT suspension samples, and (**D**) LUT-BL samples. Microscopic examination of toluidine-blue-stained pyramidal neurons in dentate gyrus/hippocampal subregions in different groups. (**E**) Normal controls, (**F**) model AD samples, (**G**) LUT suspension samples, and (H) LUT-BL samples. Black arrows = intact neurons, red arrows = damaged neurons. * statistically significant difference from normal control group and ** statistically significant difference from the SAD model group.

**Figure 11 pharmaceutics-14-00576-f011:**
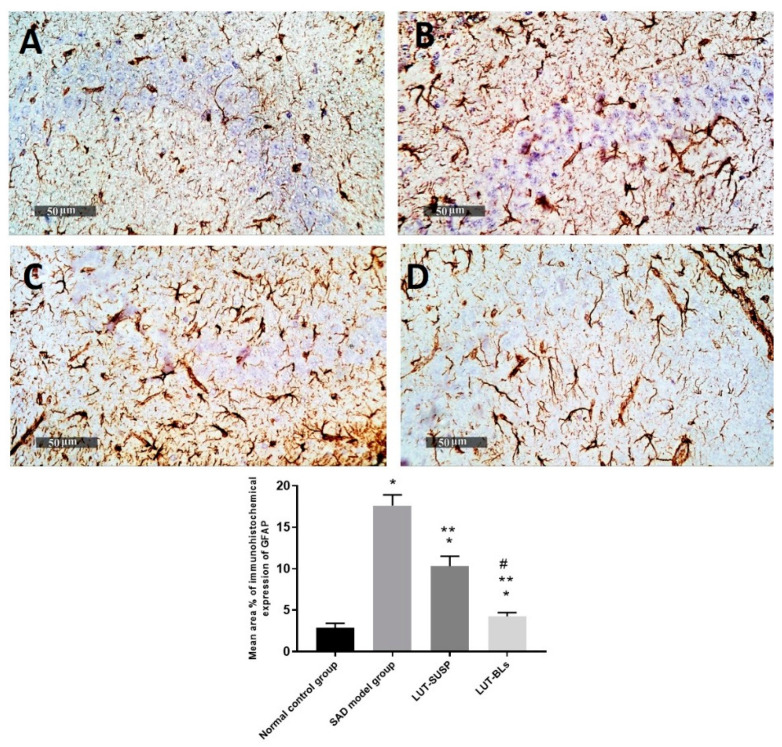
Hippocampal GFAP immunohistochemical expression levels in reactive astrocytes expressed as the mean area percentage of expression. (**A**) normal controls, (**B**) model AD samples, (**C**) LUT suspension samples, and (**D**) LUT-BL samples. Data are expressed as means ± SD. * Statistically significant difference from normal control group, ** statistically significant difference from the SAD model group, and # statistically significant difference from LUT suspension group.

**Figure 12 pharmaceutics-14-00576-f012:**
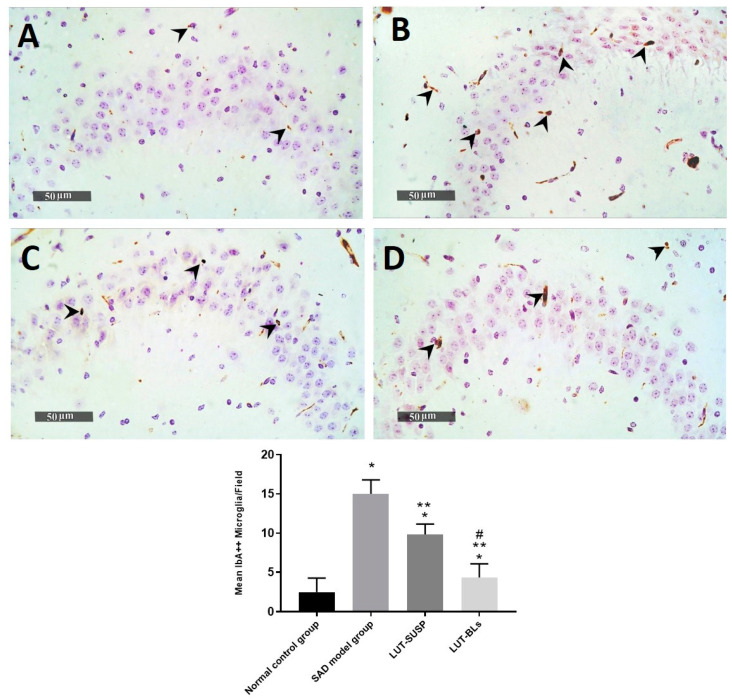
Hippocampal IbA-1/++ activated microglial cell count expressed as the mean number of IbA-1/++ cells per microscopic field. (**A**) normal controls, (**B**) AD model samples, (**C**) LUT suspension samples, and (**D**) LUT-BL samples. Data are expressed as means ± SD. * Statistically significant difference from normal control group, ** statistically significant difference from the SAD model group, and # statistically significant difference from LUT suspension group.

**Figure 13 pharmaceutics-14-00576-f013:**
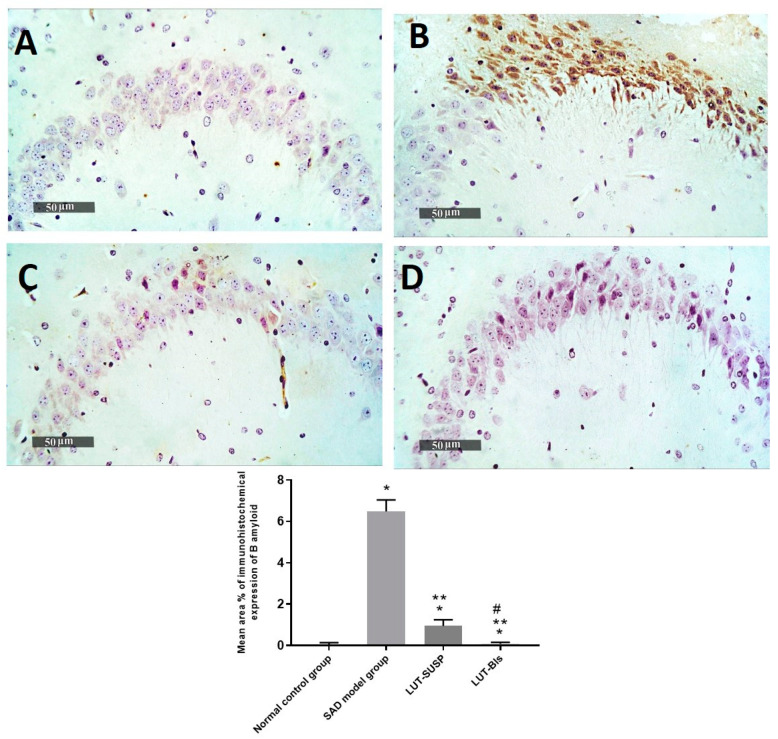
Ameliorative effect of different treatments on hippocampal intraneuronal beta amyloid immunohistochemical levels in pyramidal neurons. (**A**) Normal controls, (**B**) AD model samples, (**C**) LUT suspension samples, and (**D**) LUT-BL samples. Results were expressed as mean ± SD. * Statistically significant difference from normal control group, ** statistically significant difference from the SAD model group, and # statistically significant difference from LUT suspension group.

**Table 1 pharmaceutics-14-00576-t001:** 2^3^ full factorial design variables and constraints.

Independent Variables	Levels		
	−1	+1	Constrains
A: Surfactant concentration (mg)	100	200	In range
B: CH:LP molar ratio (*w*/*w*)	2:1	4:1	In range
C: SDC concentration (mg)	10	25	In range
Dependent variables			
R1: VS (nm)			Minimize
R2: EE (%)			Maximize
R3: ZP (mV)			Maximize
R4: Q8 h (%)			Maximize

LP: Lipoid^®^S100 ratio, CH: cholesterol, SDC: sodium deoxycholate, VS: vesicle size, %EE: % entrapment efficiency, ZP: zeta potential, and %Q8 h: % drug dissolved after 8 h.

**Table 2 pharmaceutics-14-00576-t002:** Composition and dependent variables for the developed LT-loaded bilosome formulations.

Code	A Surfactant Concentration (mg)	B CH:LP Molar Ratio (*w*/*w*)	C SDC Concentration (mg)	R1 PS (nm)	R2 EE (%)	R3 ZP (mV)	R4 Q8 h (%)
BL1	50	4	25	236.1 ± 4.21	95.1 ± 0.47	−39.0 ± 0.89	57.8 ± 1.2
BL2	50	4	10	250.0 ± 2.14	91.3 ± 0.89	−31.0 ± 0.88	42.4 ± 2.00
BL3	100	4	10	217.3 ± 1.95	83.0 ± 0.92	−33.7 ± 0.54	63.0 ± 0.97
BL4	100	2	25	153.2 ± 0.98	70.4 ± 0.77	−42.8 ± 0.24	80.0 ± 1.10
BL5	100	2	10	207.0 ± 1.57	76.0 ± 0.45	−33.0 ± 0.10	71.2 ± 1.71
BL6	100	4	25	180.0 ± 0.87	80.1 ± 0.11	−42.2 ± 0.17	70.3 ± 1.20
BL7	50	2	10	222.4 ± 1.33	87.0 ± 0.42	−32.7 ± 0.09	50.8 ± 0.72
BL8	50	2	25	190.0 ± 0.88	92.0 ± 0.11	−40.0 ± 0.16	72.1 ± 0.33

Data are expressed as mean ± SD (n = 3).

**Table 3 pharmaceutics-14-00576-t003:** 2^3^ full factorial design results.

Response	STD. Dev *	R2	Adjusted R2	Predicted R2 **	Adequate Precision ***	F-Value	*p*-Value	%CV
Y1: VS (nm)	1.06	0.9998	0.9989	0.9897	98.523	1035.8	0.0238	0.51
Y2: EE (%)	0.35	0.9997	0.9982	0.9833	66.552	637.00	0.0303	0.43
Y3: ZP (mV)	0.53	0.9982	0.9876	0.9286	23.031	93.91	0.0478	1.45
Y4: Q8 h (%)	0.95	0.9993	0.9951	0.9548	47.008	235.88	0.0398	1.51

* STD. Dev.: standard deviation; ** the predicted R^2^ is in reasonable agreement with the adjusted R^2^; *** adequate precision measures the signal-to-noise ratio (a ratio greater than 4 is desirable).

**Table 4 pharmaceutics-14-00576-t004:** Optimum levels of independent variables of the suggested LUT BL (BL4) with the expected and observed dependent variables.

Factors	Optimum Dependent Variable Levels	
**A: surfactant concentration (mg)**	100	
**B: CH: LP molar ratio (*w*/*w*)**	2	
**C: SDC concentration (mg)**	25	
**Dependent variables**	Observed value	Expected value
**R1: VS (nm)**	153.2 ±0.98	156.4 ±1.09
**R2: EE (%)**	70.4 ± 0.77	71.7 ± 0.57
**R3: ZP (mV)**	42.8 ± 0.24	43.1 ± 0.62
**R4: Q8 h (%)**	80.0 ± 1.10	81.4 ± 1.34

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
