# Peer review of "A Brain-Targeted Approach to Ameliorate Memory Disorders in a Sporadic Alzheimer’s Disease Mouse Model via Intranasal Luteolin-Loaded Nanobilosomes"

_pharmaceutics, 2022, doi:10.3390/pharmaceutics14030576_

Round 1

Reviewer 1 Report

In this work, LUT-loaded BLs was used as a novel tool to enhance LUT solubility, BBB permeability and circumvent its first-pass metabolism.

 In vivo experiments were done on an AD mouse model.

REMARKS

INTRODUCTION (note)

Please, follow the down below upgrade.

“However, many hurdles can hamper drug targeting to the brain, such as the blood-brain barrier (BBB), which blocks most neurological drugs administered either via the oral or invasive parental routes, leading to side effects as the drugs can affect non-targeted organs [5, 6]. There is a high demand for targeted and effective drug delivery to the site-of-action in the organism [https://doi.org/10.1038/s41598-021-99678-y]. Among them, non-invasive drug delivery to the brain in chronic central nervous system disorders treatment development is challenging. The olfactory route is an excellent alternative to the oral 56 or parental routes regarding brain targeting since it allows the administration of lower doses with minimum off-target side effects [7, 8].”

EXPERIMENTAL (note)

Please, provide a schematic diagram of experimental set up and provide it as a figure.

CONCLUSION (note)

In conclusion, please provide author´s future aims in this scope of investigation.

Reviewer 2 Report

In this manuscript titled “A Brain-Targeted Approach to Ameliorate Memory Disorders in a Sporadic Alzheimer’s Disease Mouse Model Via Intranasal Luteolin-Loaded Nanobilosomes,” the authors tried to evaluate the impact of intranasally delivered luteolin on the AD using bile salt-based nano-vesicles (bilosomes). During the optimization of its stability, drug release pattern, and impact, they tested several parameters like surfactant, the molar ratio of cholesterol: phospholipid, and the concentration of bile salt, associated with it. In vivo experiments were conducted on the AD mouse model via intracerebroventricular injection of 3 mg/kg streptozotocin. They have demonstrated that luteolin bilosomes improved short-term and long-term spatial memory using behavioral, biochemical markers, histological, and immune histochemistry assays on an AD mouse model, administering them a luteolin suspension or luteolin bilosomes (50 mg/kg) intranasally for 21 consecutive days. Later on, they also showed that it reduced both amyloid β aggregation and hyperphosphorylated Tau protein levels in the hippocampus. Briefly, the authors showed the efficacy and safety of the bilosomes is better than the luteolin suspension in AD treatment. This manuscript is scientifically sound. Though in my opinion, a few minor improvements in the manuscript are required.

  • On page 2, line 81, authors may include references for this statement “LUT possesses neuroprotective activity since it can suppress Aß deposition, down-regulates the expression of oxidative stress markers (by increasing glutathione levels and scavenging reactive oxygen species), and reduces pro-inflammatory mediators (NOS, COX-2, and TNF-a) levels.”

  • On page 2, line 96, the author also may mention a reference from where they adopted “A 23 full factorial design was adopted”.

  • In figure 3, it is very interesting to see the impact of the variables on the R1, R2, R3, and R4 using those polynomial equations. It will be helpful to the reader if the authors will explain it clearly. The number inside the figure is blurred (not readable).

  • In Figure 4, the figure is not well placed and the upper part is not readable. Again, some more explanation will also help readers to understand these curves.

  • I am wondering if the authors tried to scale the size of the BL4 obtained from the TEM image? Is it consistent with their particle size obtained from dynamic light scattering data? The author may comment on it in their manuscript.

  • In figure 6A, the author mentioned that “The LUT-SUSP and LUT-BS groups both displayed better short-term memory acquisition by approximately 57 % and 65 %, respectively than the STZ control group”. Whereas, the SAD model, was observed at around 50% (two-fold less than the normal one). In that case, the change is not that significant. A justification is required in the result and discussion section.

  • Although the whole result and discussion part is technically sound, the author may merge some subsections in the result and discussion part like explaining the effect of the variables on the different BLs with the optimization of the parameters. Otherwise, it sounds repetitive.

  • Authors may be careful about it. For example, in the abstract, the authors mentioned entrapment efficiency% as 70.4±0.77% but % drugs released after 8h are mentioned as 80.0% ± 1.10. The percentage sign may be placed after the standard deviation (SD). There are several typos errors throughout the manuscript.

Round 2

Reviewer 1 Report

Authors have accomplished all given remarks. Thus, I consider publication of the manuscript at current form.

In the proof version, please, correct the error remained in the titling of the last author.